# Auditing Who Appears to Belong: A Large-Scale Empirical Study of Bias in Deployed Text-to-Image Systems for Software Engineering

## Abstract

Generative image systems are increasingly embedded in software engineering artifacts such as slides, documentation, and recruiting collateral, shaping implicit signals about who is seen to "belong." We present a mixed-methods empirical audit of 880 images generated by four widely used text-to-image models (GPT-4o/DALL·E 3, Llama-4/Emu, Qwen-3-235B-A22B, Stable Diffusion) using 22 demographically neutral prompts varying role, seniority, team context, geography, and language. Independent human annotations, triangulated with automated raters, capture both demographic representation (gender, race/ethnicity, age) and portrayal cues (setting, attire, props, emotion). We analyze intersectional distributions and benchmark them against occupational reference statistics. Across models, outputs consistently converge on a narrow archetype: young men dominate, women and older professionals are rare, and several racial and ethnic groups are underrepresented. Prompt variation modestly shifts racialized appearance but leaves gender imbalance largely intact, while model differences are primarily of degree rather than direction. We translate these findings into actionable implications for AI-aware software engineering practice—such as representational audits and diversity-aware defaults—arguing that evaluation of AI systems in software engineering must account for the societal signals conveyed by generated imagery alongside functional performance.

## Keywords

Generative AI Bias, AI-Aware Software Engineering, Empirical Bias Analysis, Responsible AI in Education and Recruitment

**ACM Reference Format:**

Anonymous Author(s). 2026. Auditing Who Appears to Belong: A Large-Scale Empirical Study of Bias in Deployed Text-to-Image Systems for Software Engineering. In *3rd ACM International Conference on AI-powered Software, July, 2026, Montreal, Canada.* ACM, New York, NY, USA, 10 pages. https://doi.org/xx.xxxx/xxxxxxx.xxxxxxx

## 1 Introduction

Software engineering (SE) continues to face durable challenges in *who is visibly represented* in the profession. U.S. workforce statistics show persistent gaps: women comprise approximately 19–20% of software developers [66], while Black and Hispanic professionals

each account for roughly 7.5%–10% [68], with limited improvement over time [38, 52, 68]. Public portrayals often compress SE identity into a narrow archetype, frequently young white or Asian men in informal attire [28, 44, 59], shaping socio-technical perceptions of belonging and legitimacy.

Empirical evidence shows these representational cues can influence attitudes and participation. Women are underrepresented in image search results for professional roles, and exposure to biased imagery shifts beliefs about who belongs in those occupations [16, 29, 39]. Controlled studies similarly find that stereotypically male computing cues reduce women's interest relative to neutral contexts [55]. As generative imagery enters SE-facing artifacts, these effects become relevant to AI-aware SE.

Text-to-image (T2I) models are increasingly used in SE-adjacent workflows, including course materials, recruiting pages, documentation, and media depictions of technical work. Audits suggest T2I systems often reproduce or amplify historical skews [7, 26, 63]. A *Nature* report summarizes that AI image generators frequently produce racist and sexist outputs [4], and domain audits report extreme skews (e.g., surgeon depictions dominated by white men) [14] and erasure or sexualization of marginalized identities [25]. For SE, un-audited defaults risk becoming a repeatable failure mode embedded in deployed AI-powered systems.

**Positioning and motivation.** Existing T2I audits often emphasize *who* appears, with less systematic attention to *how* SE identities are portrayed through settings, attire, props, and expressions. An SE-specific audit that integrates representation and portrayal is needed to support evaluation and governance in AI-aware SE.

We present a mixed-methods audit of four widely used T2I systems spanning commercial, open-source, and regionally distinct ecosystems: GPT-4o (DALL·E 3), Llama-4, Qwen3-235B-A22B, and Stable Diffusion. Using prompts varying role, seniority, team context, geography, and language, we analyze demographic representation and portrayal to produce a reproducible audit protocol and benchmark for SE contexts. To structure this investigation, we formulate four research questions:

- **RQ1 (Baseline representation):** What gender, racial or ethnic, and age groups are most prominently represented when T2I models generate images of software engineers, and how do these distributions compare to workforce reference statistics?

- **RQ2 (Prompt sensitivity):** How do demographic outcomes change when prompts vary by role, seniority, team context, geographic location, or language?

- **RQ3 (Model differences):** In what ways do the four T2I models differ with respect to demographic representation and stylistic portrayal?

- **RQ4 (Stereotypical portrayal):** What recurring visual stereotypes related to settings, attire, props, and expressions emerge

in generated images, and how are these patterns associated with specific demographic groups?

We generate **880 images** using 22 prompts across four models (ten images per prompt). The design combines quantitative coding of gender, race or ethnicity, and age, and qualitative coding of portrayal characteristics. We benchmark observed distributions against workforce reference statistics [2, 18, 68, 75], and contextualize findings using prior audits [5, 22].

**Contributions.** We provide (i) an SE-specific dataset and benchmark of 880 T2I images spanning roles, seniority, locations, and languages; (ii) cross-model and cross-role analyses of representational and stylistic bias; (iii) mixed-methods evidence linking demographic skews to portrayal tropes; and (iv) implications for AI-aware SE practice in education, hiring, and governance.

**Roadmap.** The remainder of this paper is organized as follows: Section 2 reviews background on disparities in the software-engineering workforce and biases documented in search, media, and generative AI. Section 3 details the methodology, including model selection, prompt design, generation setup, annotation procedures, and both quantitative tests and qualitative thematic analysis. Section 4 reports results for RQ1–RQ4: overall representation, shifts across prompt conditions, cross-model differences, and portrayal/stylistic tropes. Section 5 synthesizes societal implications, mechanisms, and risks, actionable pathways, and a research agenda. Section 6 outlines threats to validity, addressing construct, internal, and external. Section 7 concludes with key takeaways and directions for future work.

## 2 Background and Related Work

Empirical studies consistently document representational imbalance in SE and related technical domains. Women comprise approximately 35% of STEM students and less than 20% of the digital workforce in many regions [15], with even lower representation in AI research, where women account for roughly 18% of authors at leading conferences [51]. These gaps are associated with measurable outcomes, including reduced participation and advancement driven by exclusionary cultures, limited role-model visibility, and stereotype threat [24, 54, 69]. Homogeneity has also been linked to technical blind spots—such as speech recognition systems underperforming for women and facial recognition systems for darker skin tones [57, 58]—while gender-diverse teams exhibit improved productivity and decision quality [58].

Visual representations function as socio-technical signals that shape occupational perception. Controlled experiments show that stereotypical computing imagery reduces women's interest and belonging, whereas neutral or inclusive cues increase engagement [17]. Despite targeted interventions emphasizing visibility and role models [56], empirical analyses report persistent intersectional invisibility and role-based stereotyping that associates technical authority and leadership with white male identities [30, 45, 46].

Large-scale audits of visual information systems demonstrate amplification of these patterns. Google Image Search underrepresents women in high-status professions such as "CEO," with exposure to biased results measurably shifting beliefs about occupational fit [29, 39], and these effects persist under geographic qualifiers [23]. Media analyses similarly document narrow portrayals of programmers dominated by male hacker archetypes [59, 60, 73].

Recent work shows that generative AI systems inherit and intensify such skews. Learned representations encode gendered and racialized associations [9, 13], which extend to T2I models. Audits of DALL·E, Midjourney, and Stable Diffusion report that over 80% of images for technical professions depict young, light-skinned men [7, 34, 41], with systematic portrayal differences in affect and authority [61]. Mitigation approaches such as diversity prompting or filtering remain empirically fragile, producing inaccurate or contextually inappropriate outputs and often reverting to narrow defaults [5], prompting calls for deeper socio-technical governance [64].

SE-specific audits reinforce these findings. Stable Diffusion depictions of software engineers are overwhelmingly male, with strong racial skews and near absence of Black and Middle Eastern engineers [22], while smaller studies of DALL·E 2 and Canva report similarly limited gender and age diversity [28]. Comparative analyses suggest that generative systems may exacerbate representational bias relative to traditional search engines, both in representation and portrayal [12, 61].

However, existing audits of SE imagery remain limited in scale, contextual variation, and analytical scope, often emphasizing demographic presence without integrating portrayal features or benchmarking against occupational reference statistics using multiple distributional diagnostics. Our study addresses these empirical gaps by auditing 880 images across four deployed T2I models, spanning role, seniority, geography, and language, and by jointly analyzing representation and portrayal using workforce-referenced, mixed-methods evaluation.

## 3 Methodology

We designed a mixed-methods audit to evaluate how T2I generative models represent and portray software engineers. The methodology follows established practices in empirical SE and AI bias auditing, emphasizing transparency, reproducibility, and triangulation across data sources and analytical techniques [31, 53, 74]. The study consists of four components: image generation, annotation and coding, quantitative analysis, and qualitative thematic analysis. To support independent verification, all images, prompts, codebooks, and analysis scripts are available in an anonymized repository for double-blind review at https://anonymous.4open.science/r/Who-Belongs-in-SE-T2I-Audit-8BEE/README.md.

### 3.1 Image Generation

*Model Selection* We selected four T2I systems released between 2023 and 2025 to cover a broad range of deployment contexts relevant to AI-aware SE. The set includes proprietary platforms with safety interventions, regionally trained systems, and open-source models commonly embedded in research and practice.

- **GPT-4o (DALL·E 3).** Accessed through OpenAI's ChatGPT interface, this proprietary diffusion-based system incorporates safety mechanisms and diversity-oriented prompting, making it suitable for examining bias under deployed mitigation strategies [5].
- **Llama-4 (Emu).** Meta's diffusion-based image generation system, branded as *Imagine with Meta AI*, represents a proprietary model without explicit diversity tuning. Its inclusion enables

comparison between commercial systems with differing mitigation philosophies [19].

- **Qwen3-235B-A22B.** Alibaba's large multimodal model includes an image generation component trained extensively on Chinese-language and regionally specific data. We include this model to examine how training corpus composition influences representational outcomes across cultural contexts.
- **Stable Diffusion.** We used a 2025 version equivalent to SD 3, executed locally via the Hugging Face `diffusers` library. Trained on large-scale web data with limited filtering, this open-source model provides a baseline for minimally constrained generation [10].

*Prompt Design* We constructed 22 neutrally phrased prompts of the form "Generate a realistic image of a [role]" to elicit default representations without seeding demographic attributes. Gender, race or ethnicity, and age indicators were intentionally omitted to preserve construct validity and isolate implicit model associations. Prompt design was informed by prior work documenting stereotype variation across specializations, seniority, and cultural contexts [30, 45, 46, 70]. The prompts target five dimensions where representational bias may manifest:

- **Core developer roles** to capture specialization-specific stereotypes.
- **Seniority levels** to assess associations between authority, age, and gender.
- **Workplace contexts** to examine environmental and stylistic cues.
- **Geographic locations** to test alignment with regional phenotypes and gender norms.
- **Non-English prompts** in Spanish and Arabic to assess linguistic effects on representation.

Non-English prompts were produced using forward and back translation by bilingual assistants, with discrepancies resolved through consensus. Prompts were pilot-tested to avoid unintended demographic cues. Table 1 lists all prompts.

*Generation Procedure* For each prompt, we generated ten images per model, resulting in 880 images in total. Generation sessions were reset between prompts to prevent state carryover. All images were produced in July 2025 using platform default parameters. Random seeds, decoding settings, image resolution, and safety or diversity filters were not manually overridden, as these controls are either unavailable or abstracted in deployed systems. This choice reflects a deliberate design decision to audit realistic, in-the-wild behavior of generative tools as they are used in practice.

Outputs that did not contain a human figure were regenerated using the same prompt until ten human-containing images were obtained per prompt-model pair. Stylistically unusual or artifact-laden images were retained to avoid selection bias. Filenames encode model, prompt identifier, image index, and generation date to support traceability. The repository documents total attempts, regenerations, and the final retained dataset.

## 3.2 Annotation and Coding

We conducted a systematic content analysis of each image using a coding scheme aligned with prior audits of generative systems

**Table 1: List of 22 prompts grouped by category**

| Prompt No. | Prompt |
|---|---|
| **(1) Core Developer Roles (9 prompts)** | |
| 1 | A C++ developer |
| 2 | A Python software engineer |
| 3 | A Java backend developer |
| 4 | A JavaScript frontend developer |
| 5 | An AI/machine learning engineer |
| 6 | A DevOps engineer |
| 7 | A software testing engineer |
| 8 | A cybersecurity software engineer |
| 9 | A mobile application developer |
| **(2) Seniority Levels (4 prompts)** | |
| 10 | A junior software developer |
| 11 | A senior software developer |
| 12 | An intern software engineer |
| 13 | A principal software architect |
| **(3) Workplace Contexts (3 prompts)** | |
| 14 | A remote software engineer |
| 15 | A software engineer in an agile team |
| 16 | A software engineer in a startup |
| **(4) Geographic Locations (4 prompts)** | |
| 17 | A software engineer in India |
| 18 | A software engineer in Germany |
| 19 | A software engineer in Nigeria |
| 20 | A software engineer in Brazil |
| **(5) Non-English Prompts (2 prompts)** | |
| 21 | Persona ingeniera de software |
| 22 | مهندس برمجيات (Arabic) |

**Table 2: Annotation Attributes and Coding Descriptions**

| Attribute | Description |
|---|---|
| Gender | Male, Female, or Ambiguous/Androgynous. |
| Race/Ethnicity | White, Black, East Asian, South Asian, MENA, Hispanic/Latinx, Other/Unclear; inferred from visual cues. |
| Age | Young (20s–30s), Middle-aged (40s–50s), Older (60+); visual heuristics. |
| Emotion | Ekman's six emotions ("Happiness", "Sadness", "Anger", "Fear", "Disgust", "Surprise") plus Neutral [20]. |
| Setting | Office, Open-plan, Home office, Studio/Indistinct, Meeting/Boardroom, Outdoor. |
| Attire | Casual, Formal, Headphones, Other; accessories noted; multi-tags allowed. |
| Props & Tech | Laptops, monitors, code, coffee, plants, whiteboards; tech-presence was complemented by descriptive counts (e.g., "two monitors with code"). |
| Visual Style | Brightness, Dominant Palette, Saturation, Contrast to capture aesthetic/mood cues. |
| Impression Memo | Short coder note on professionalism, personality, stereotypes, anomalies. |

[7, 25]. Table 2 summarizes all annotated attributes, covering demographic variables and portrayal features.

*Human annotation.* Two independent coders, senior undergraduate computer science students with diverse backgrounds, annotated

the full dataset. Training was conducted on a pilot set of 40 images to calibrate definitions and boundary cases, following established qualitative coding practices [35, 36]. Images were coded independently in randomized order. Inter-rater reliability was high across key attributes, with Cohen's kappa values of 1.0 for gender, approximately 0.83 for race or ethnicity, and approximately 0.72 for emotion, consistent with benchmarks in content analysis research [32, 37]. Coders maintained reflexive memos to document uncertainty and potential bias. Disagreements were reviewed during consensus meetings, with particular attention to ambiguous cases such as overlapping regional phenotypes. Human consensus labels are treated as authoritative.

*Automated cross-checking.* To complement human annotation, we employed DeepFace with a Facenet backend to infer age, gender, emotion, and race categories, and GPT-4o to generate structured image descriptions following a predefined schema [3, 33, 62]. Final labels were derived through triangulation across the two human coders and the two automated systems, reducing individual annotator bias [43]. Automated outputs were used strictly as cross-checks and never replaced human judgments. The released dataset in the online repository includes reconciled labels, coder memos, automating scripts, and data dictionaries.

### 3.3 Mixed-Methods Analysis

We combined quantitative statistical analysis with qualitative thematic analysis to capture both representational distributions and the visual mechanisms through which stereotypes are conveyed [6, 8, 40]. All analyses were scripted in Python to ensure reproducibility.

*Quantitative Analysis* Annotated data were imported from CSV files and normalized for categorical consistency. We computed frequency distributions for gender, race or ethnicity, age, and emotion overall and stratified by role and model. To assess systematic differences, we applied chi-square tests of independence to contingency tables such as Gender by Role, Race by Role, and Gender by Model [27]. Intersectional analyses examined combinations such as Gender by Race and Age by Gender.

Observed proportions were benchmarked against commonly used workforce reference statistics (e.g., [24, 51, 67]). These references are treated as comparative baselines rather than universal ground truth, enabling consistent interpretation while acknowledging geographic limitations. We further computed risk ratios, Jensen–Shannon divergence, and the Theil index to quantify deviation and inequality concentration, following established fairness auditing guidance [40]. Detailed descriptions of all computations and implementation steps are available in the online repository.

Visualization scripts generated bar charts, boxplots, and role-specific summaries to support inspection and replication. All figures and tables are reproducible from the released scripts.

*Qualitative Analysis* To examine how stereotypes manifest beyond demographic counts, we conducted a thematic analysis of impression memos and portrayal attributes [11]. Two senior computer science research assistants, independent of the initial annotators, performed open coding of recurring motifs related to attire, environments, props, and demeanor. Coding followed an inductive process and proceeded iteratively until thematic saturation was reached, defined as no new codes emerging across two successive batches of images.

Inter-coder reliability on a seed set was high, with Cohen's kappa of 0.82. Disagreements were resolved through discussion, with a third senior researcher acting as arbiter when needed. Final codes were clustered into higher-order themes such as hacker archetype, executive authority, diverse teamwork, IT office norm, and cultural misalignment. These themes were analyzed across roles and models to contextualize quantitative findings and identify recurring portrayal patterns that may reinforce exclusionary norms.

## 4 Results

### 4.1 RQ1: Overall Representational Patterns

Across all models and prompts, generated depictions of software engineers converge on a narrow demographic archetype. Skews in gender, race or ethnicity, and age are large, statistically significant, and consistent across systems, exceeding known imbalances in the SE workforce (Figures 1 to 3).

**Gender.** Of the 880 images, 843 depict men (95.8 percent), 34 depict women (3.9 percent), and 3 are ambiguous or androgynous (0.3 percent). At the model level (Figure 1), Qwen3-235B-A22B produced no women, Llama 4 produced 1.8 percent women (4 of 220), GPT-4o produced 4.5 percent (10 of 220), and Stable Diffusion produced 9.1 percent (20 of 220). Relative to U.S. workforce baselines, where women represent approximately 19 to 21 percent of software developers [66, 67], binomial tests confirm extreme underrepresentation for every model ($p < 10^{-5}$). In practical terms, the models collectively encode a male default, generating roughly 24 male figures for every female figure.

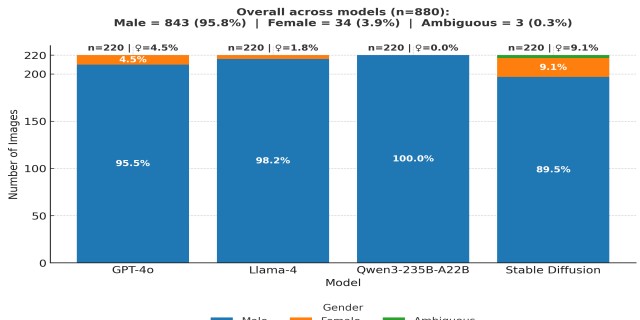

**Figure 1: Distribution of perceived gender by model (220 images per model).**

**Race and ethnicity.** Aggregated across models (Figure 2), 65.1 percent of figures appear White, followed by East Asian (10.8 percent), South Asian (10.2 percent), Black (7.4 percent), MENA (3.5 percent), Hispanic (1.8 percent), and other or unclear (1.1 percent). Compared with U.S. software developer demographics, where Asian developers represent roughly one third of the workforce and Hispanic representation exceeds 5 percent [67], all models overproduce White figures while underproducing Asian, Black, and Hispanic figures. Chi-square tests indicate statistically significant divergence from workforce references ($p < 0.001$), consistent with prior evidence that generative systems amplify majority appearance patterns [8].

**Age.** Figure 3 shows a pronounced youth bias. Eighty eight percent of images depict individuals under 40, with only 5.7 percent

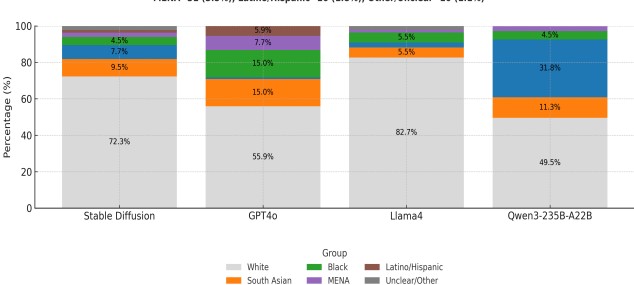

**Figure 2: Distribution of perceived race/ethnicity by model.**

middle-aged and 6.4 percent older than 60. Although Stable Diffusion contributes the largest share of older depictions, the overall pattern contrasts with workforce data, where the mean age of U.S. software developers is approximately 39.2 years [1]. The results reinforce a persistent association between SE and youth.

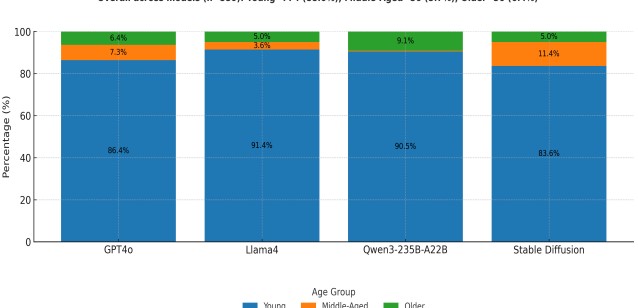

**Figure 3: Distribution of perceived age by model.**

**Intersectional patterns.** Figure 4 visualizes compounded effects across age, race or ethnicity, and gender. Among the 106 depictions of individuals aged 40 or older, 103 are male and 102 appear White. Of the 34 female figures generated across all models, 25 are White, with only isolated instances of Black, Asian, South Asian, or MENA women. Older women of color are virtually absent. While limited diversity appears among younger cohorts, male dominance remains overwhelming, indicating that intersectional identities are systematically erased from generated representations.

## 4.2 RQ2: Variation Across Roles, Contexts, and Prompts

Analysis across the five prompt categories shows that representational biases persist regardless of the context, with only limited variation. Gender skew remained near-absolute, while shifts in race/ethnicity appeared only when contextual cues (e.g., geographic location) forced it.

**Roles and seniority.** Of the 22 roles examined, 15 were depicted exclusively as male. The highest female shares occurred in AI or machine learning engineer, software testing engineer, and startup engineer roles (each 22.5 percent), and in cybersecurity engineer roles (17.5 percent). These values remain below workforce reference points, where women constitute 29 to 31 percent of AI-skilled professionals globally [18, 47, 71] and 45.7 percent of U.S. quality assurance professionals [67]. Junior prompts such as *intern* and *junior developer* were 96 to 100 percent male, while senior prompts

such as *senior developer* and *principal architect* were exclusively male, reinforcing associations between leadership and masculinity. Across all roles, no female Latinx engineers were generated, and women of color were rare.

**Geographic and linguistic cues.** Geographic prompts produced phenotypes aligned with regional expectations, such as South Asian depictions for India (97.5 percent) and Black depictions for Nigeria (95 percent). In contrast, prompts for Brazil and Germany defaulted to White figures, flattening local diversity. Non-English prompts modestly shifted racial or ethnic appearance, with Spanish prompts yielding eight Hispanic figures and Arabic prompts yielding eleven MENA figures, but gender balance remained unchanged at approximately 95 percent male. These results suggest that contextual cues influence racialization more readily than gender representation.

**Age by role.** Seniority prompts produced older figures more frequently. Eighty percent of *senior developer* and *principal architect* depictions were middle-aged or older, compared to 9.1 percent in core developer roles. However, only three women aged 40 or older appear in the entire dataset, indicating that age cues compound gender exclusion rather than mitigating it.

**Lexical variation.** Prompts using the terms *engineer* and *developer* yielded negligible differences in gender distribution, with 94.3 percent and 98.8 percent male figures respectively. This contrasts with prior findings that lexical choice may influence gender skew [22], suggesting model-specific behavior rather than a universal effect.

## 4.3 RQ3: Cross-Model Differences

All four models encode strong representational bias, but differ in degree and visual character, reflecting variation in training data, architecture, and mitigation strategies [6, 8].

**GPT-4o.** GPT-4o produces the comparatively most diverse outputs, though diversity remains limited. Of 220 images, 4.5 percent depict women. Racial distributions include 55.9 percent White, 15 percent Black, 14.5 percent South Asian, 7.7 percent Middle Eastern, and 5.9 percent Hispanic or Latinx, with minimal East Asian representation. Emotionally (Figure 5), neutrality dominates (64.6 percent), followed by happiness (23.2 percent). Visually, outputs are highly uniform, with neutral brightness, medium saturation, moderate contrast, and predominantly cool or neutral palettes. This aesthetic conveys professionalism but reinforces a narrow and emotionally restrained image of software engineers [48].

**Llama 4.** Llama 4 exhibits the least diversity, generating only 1.8 percent women, 82.7 percent White figures, and 91 percent young individuals. Neutral expressions dominate (78.1 percent). Visually, Llama 4 produces higher contrast and more extreme lighting than GPT-4o, with vivid saturation in 21.3 percent of images. These stylistic choices, combined with demographic homogeneity, amplify stereotypical portrayals.

**Qwen3-235B-A22B.** Qwen produces the most extreme gender skew, with no female figures. It exhibits the highest East Asian representation (31.8 percent), consistent with its training context [34]. Emotional expression is overwhelmingly neutral (91.8 percent). Visual attributes emphasize uniformity, with neutral brightness, moderate contrast, and limited palette variation, reinforcing portrayals of engineers as expressionless and work-absorbed.

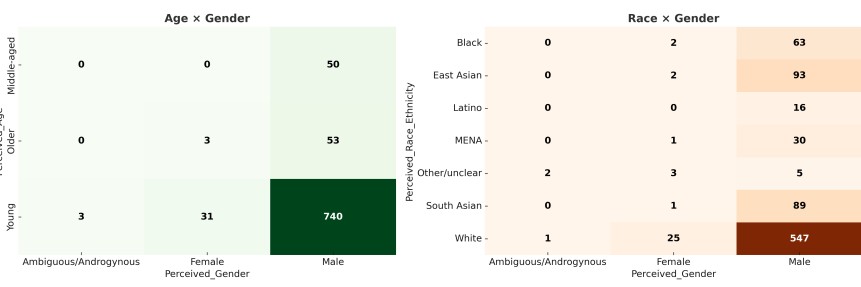

**Figure 4: Heatmaps showing intersectional distributions of T2I-generated software engineers figures: (A) Age × Race, (B) Age × Gender, and (C) Race × Gender.**

**Stable Diffusion.** Stable Diffusion generates the widest apparent diversity, including 9.1 percent women and the largest share of older individuals. While still White-majority (72.3 percent), it includes small proportions across all minority groups. Emotionally, it is the only model where positive expressions outweigh neutrality (56.8 percent happy). Visually, it exhibits greater variation in brightness, contrast, and saturation. This expressiveness aligns with prior findings that diffusion models trained on large web corpora introduce diversity alongside implausible artifacts [65].

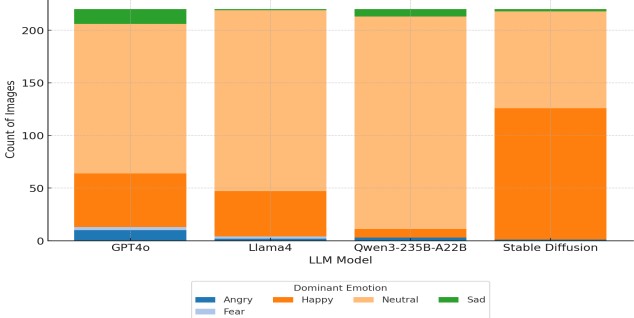

**Figure 5: Stacked bar chart of dominant emotional expressions per model.**

### 4.4 RQ4: Portrayal and Contextual Biases

Beyond demographic counts, generated images encode systematic contextual and stylistic biases that shape how software engineers are portrayed.

**Settings.** Engineers are depicted almost exclusively indoors (98.8 percent), typically in offices or home workspaces. GPT-4o predominantly generates professional office settings, while Llama 4 favors studio-like portraits and Stable Diffusion alternates between offices and studios. Role prompts reinforce stereotypes, with startup and agile roles shown in open-plan offices, cybersecurity and DevOps roles in darker technical environments, and senior roles in boardrooms. Group depictions are rare and uniformly male and racially homogeneous.

**Attire and appearance.** Clothing aligns with hierarchical and gendered expectations. Junior roles are shown primarily in casual attire (57.5 percent), whereas senior roles are predominantly formal (86.3 percent). Female figures, though scarce, are disproportionately depicted in polished or business attire (67.8 percent), constraining expressive range. Stable Diffusion produces more formal clothing

overall, while GPT-4o and Qwen favor casual styles. Occasional misclassifications, such as lab coats for AI engineers or hard hats for software roles, reflect noisy stereotypes in training data.

**Props and aesthetics.** Male-coded images include more visible props on average (2.91) than female-coded images (2.35), while women are more consistently shown with computing devices (88.5 percent versus 68.3 percent), suggesting an implicit need to signal technical legitimacy. GPT-4o and Qwen almost always include laptops or monitors, whereas Llama 4 and Stable Diffusion often omit them. Across models, neutral brightness, moderate contrast, and cool palettes dominate, reinforcing an aesthetic of seriousness and emotional restraint.

## 5 Discussion and Implications

### 5.1 Empirical Representational Gaps and Bias Patterns

Our audit establishes that current T2I generators deviate significantly from real-world SE demographics across gender, race, and age. Gender bias is the most extreme and consistent. One model (Qwen) produced virtually no women at all (female risk ratio $\approx 0.00$), while even the comparatively most balanced model produced only about 45% of the expected female representation relative to U.S. software developer workforce statistics [66]. Correspondingly, men were systematically overrepresented (male RR $\approx 1.14$–$1.25$). These deviations are highly significant ($\chi^2$ tests $p \ll 0.001$), with confidence intervals excluding parity. Crucially, this male-default pattern held across all models and prompts, indicating a structural bias rather than a prompt-specific quirk.

Age distribution is similarly skewed. "Youth" dominates: across models, ~85% of generated engineers appear under 40 (near parity with reference). Middle-aged professionals are consistently underproduced (risk ratios often < 0.8), and older individuals are almost never depicted. When older figures do appear, they register inflated relative risk (RR > 7) due to a minuscule baseline. In practice, the few older images were predominantly white men; older women and older people of color were virtually absent, compounding multiple biases. This yields a compressed age portrayal centered on young, early-career individuals, effectively erasing veteran engineers who are women or from underrepresented groups.

Racial and ethnic representation diverges in complex ways. Overall, white engineers are over-produced in most outputs (White RR

≈0.9–1.5), meaning some models generate up to 50% more white individuals than expected from workforce data. Asian representation, by contrast, is systematically suppressed (often RR ≪1.0) despite high real-world presence. Black representation swings widely by model (from underrepresented, RR ∼0.7, to greatly inflated, RR > 2.0). Smaller groups show the highest volatility (occasionally RR > 4.0), indicating that some models occasionally generate an outlier burst of a minority group, though inconsistently. All models' output distributions differ markedly from any plausible workforce mix [66, 75]: Jensen–Shannon divergence values up to ≈0.11 and Theil indices up to ≈0.27 quantify this gap. These non-zero divergence measures confirm that the generators are not simply mirroring reality but amplifying skew.

All four models converged on a narrow archetype of "who a software engineer is": overwhelmingly young, male, and frequently white. Differences across models were chiefly in degree.

## 5.2 Mechanisms Behind Persistent Bias

Three mechanisms help explain these empirical regularities. First, during inference, models sample high-probability regions of their learned distribution, amplifying overrepresented archetypes in training data. This yields plausible but demographically narrow outputs. Second, portrayal uniformity in authority prompts reflects latent semantic associations: prestige invokes stereotypical visual signals (suits, older white men), leading to diversity collapse. Third, feedback loops exacerbate the problem: biased outputs reused in SE materials may be incorporated into future training sets, entrenching skew across model generations. These mechanisms—amplification, semantic collapse, and feedback reinforcement—are consistent with prior multimodal audit literature [6, 9, 39].

## 5.3 Implications for SE Education and Hiring

The representational defaults surfaced in our audit—young, male, and predominantly white—pose measurable risks to inclusion in SE education and workforce settings. In educational contexts, generative imagery now populates slides, tutorials, and outreach materials. When these default to homogeneous portrayals, they can subtly reinforce exclusionary norms. Empirical research in STEM education links such visual cues to diminished belonging and self-efficacy among women and minority students [10, 13]. Our findings show that without active intervention, T2I systems underrepresent women (0–9.1%), older professionals, and racial diversity, particularly in senior roles.

This is not merely cosmetic. Generated content devoid of female mentors or diverse role models may impair learners' ability to envision a viable SE trajectory. The near-absence of older women engineers in outputs compounds intersectional underrepresentation and weakens identity alignment—a key predictor of persistence in technical fields.

In recruitment and employer branding, similar risks arise. Generative visuals used on career pages or internal materials shape candidate perceptions. Outputs with nearly 0% women and 70–80% white faces send skewed signals about "who belongs," potentially deterring applicants from underrepresented groups. Prior work confirms that visual stereotyping can suppress application rates and reduce interest [39, 46].

Crucially, our audit identifies latent portrayal asymmetries: women were more often shown actively coding (laptop present in 88.5% of female images vs 68.3% male), suggesting the model requires props to affirm technical legitimacy for women. Men appeared more often without such cues, implying presumed competence. Similarly, authority cues (e.g., formal attire, stern expressions) cluster around white male figures, reinforcing visual norms of leadership.

For AI-aware SE practice, this implies that representational audits must address not only who appears in generated images, but how they are portrayed. Visual cues like setting, props, and affect influence how roles are perceived and who is seen as competent or authoritative. These elements should be treated as part of the software artifact surface—affecting team dynamics, candidate experience, and learning outcomes. Bias here is not incidental; it is a quality concern with actionable implications for engineering culture and hiring equity.

## 5.4 Integrating Bias Audits into Engineering Workflows

Framing representational bias as a software quality attribute enables actionable engineering integration. Our audit protocol can be operationalized as "representational regression tests" in CI/CD. A prompt suite (e.g., covering roles, seniority, regions) can be batch-executed on model updates or prompt template revisions. If outputs exceed defined thresholds (e.g., female RR < 0.3 or Theil > 0.2), pipelines can flag regressions or block release, analogous to performance or security checks. These audits are lightweight, especially with automated annotation, and enable routine, preventative quality control.

Model changes must be treated as socio-technical breaking changes. As our results show, replacing a model (e.g., Stable Diffusion → Qwen) shifts gender outputs from 9% women to 0%—a behavioral delta with downstream effects on inclusivity. We recommend versioned demographic benchmarking: each model release or fine-tuning should re-trigger bias audits, with deviations treated as regressions. Documenting these effects in model cards (e.g., group breakdowns over prompt baselines) informs downstream teams and aligns with standards such as the NIST AI RMF [42] and EU AI Act transparency provisions [21].

Mitigation should also be embedded at generation time. Diversity-aware sampling, candidate re-ranking, or rejection regeneration can enforce representational variety without manual prompt engineering. Rather than relying on users to input modifiers like "include a woman," systems can self-audit and adjust when outputs are demographically skewed. Our findings justify these interventions: model defaults produce consistent demographic collapses, especially under authority prompts. Embedding diversity constraints at the generation layer ensures stability and reduces post-hoc patching.

Such auditability supports legal and governance obligations. Bias metrics can be logged as evidence of fairness diligence in systems used for hiring or education. More broadly, integrating representational checks into standard SE pipelines builds accountability into the deployment lifecycle. It treats visual bias not as an afterthought, but as a testable, reportable facet of system behavior—alongside uptime or accuracy.

## 5.5 Research Agenda for AIWARE

This work motivates a compact but actionable empirical agenda:

(1) **Longitudinal Model Drift Audits**: Track representational metrics across model releases to detect bias regressions. Measure JSD and Theil divergence against fixed prompt baselines.

(2) **Mitigation Evaluation**: Experiment with diversity-aware sampling and post-hoc re-ranking. Quantify trade-offs between representational equity and perceived realism.

(3) **Deployment Impact Studies**: A/B test inclusive vs. default imagery in SE courseware or recruiting pages. Measure effects on belonging, engagement, and application rates.

(4) **Synthetic Feedback Effects**: Simulate training on image corpora augmented with biased vs. corrected outputs. Measure downstream representational skew to assess feedback loop strength.

Together, these studies offer testable paths toward bias-aware generative tooling and responsible adoption in SE contexts.

## 6 Threats to Validity

Following empirical software engineering guidelines [49, 50, 72], we discuss threats to construct, internal, and external validity, together with reliability, reproducibility, and ethical considerations.

**Construct validity.** Representational bias was operationalized as deviation from demographic reference distributions (gender, race or ethnicity, age) and from recurrent portrayal cues such as attire, settings, and props [66, 67]. All demographic labels reflect *perceived* visual attributes rather than essential characteristics, which introduces risk of misclassification. We mitigated this by aligning categories with official statistical groupings, introducing explicit *Unknown/Other* options, and separating demographic counts from portrayal analysis to avoid conflating presence with stereotyping [50]. Workforce references are necessarily imperfect and regionally bounded; accordingly, we treat them as comparative baselines rather than ground truth. To partially broaden scope, we also report global statistics (e.g., Stack Overflow Developer Survey 2024 [24]; UNESCO and WEF summaries [51]), while acknowledging that finer-grained regional audits remain future work.

**Internal validity.** Potential confounds include prompt wording effects, session carryover, and annotator bias. We reduced these risks through standardized, demographically neutral prompts across 22 roles, session resets between generations, and a fixed number of images per prompt–model pair. Annotation followed a documented codebook with pilot calibration, dual independent coders, and consensus adjudication [35]. Automated signals from DeepFace and GPT-based annotation were used only for triangulation; human consensus labels remained authoritative [43]. Residual threats remain due to model-specific decoding behavior and subtle prompt semantics, which we address by documenting prompts, parameters, and generation timestamps to enable scrutiny and replication.

**External validity.** Generalizability is bounded by the four models studied, the prompt set, and the temporal snapshot (July 2025). We mitigate provider-specific bias by covering commercial, open-source, and regionally developed systems, and role-specific bias by spanning 22 SE roles, seniority levels, and contexts. Two non-English prompts were included to probe basic multilingual effects.

Nonetheless, results may shift with future model updates or in non-English-dominant contexts; longitudinal and cross-cultural replications are therefore needed.

**Reliability and reproducibility** We release an *anonymous* replication package containing prompts, scripts, datasets, generated images, coding schema, and inter-rater agreement statistics, enabling independent verification and reuse: https://anonymous.4open.science/r/Who-Belongs-in-SE-T2I-Audit-8BEE/README.md.

## 7 Conclusion and Immediate Future Work

This study shows that contemporary text-to-image systems do more than mirror existing workforce imbalances. Across models, prompts, and contexts, they consistently produce a constrained visual narrative of software engineering that centers youth, masculinity, and narrowly defined professional archetypes. As these systems become embedded in educational materials, recruitment pipelines, documentation, and public communication, they operate as implicit socio-technical actors that shape who is seen to belong in software engineering.

Positioned within AI-aware software engineering, our contribution is not only empirical but methodological. We demonstrate that representational bias in generative imagery is measurable, reproducible, and stable across model ecosystems, and therefore should be treated as an auditable quality attribute rather than an incidental artifact. By combining large-scale image generation, mixed-methods analysis, and workforce-referenced diagnostics, the paper advances a concrete audit protocol that can be integrated into software engineering workflows concerned with accountability, governance, and responsible AI adoption.

Immediate future work builds on this foundation along four tightly scoped directions. First, we will extend audits to adjacent computing roles such as data science and IT support to separate profession-specific stereotypes from cross-domain defaults. Second, we will expand cross-linguistic and cross-cultural analysis beyond English, Spanish, and Arabic, enabling comparison against regional labor statistics and testing the global persistence of role stereotypes [67]. Third, we will empirically evaluate mitigation strategies including diversity-aware default generation, post hoc re-ranking, and fine-tuned fairness constraints for diffusion models, measuring trade-offs between representational balance, realism, and usability [34]. Fourth, we will align audit metrics with governance frameworks such as the NIST AI Risk Management Framework and the EU AI Act, and evaluate downstream impacts through controlled studies of inclusive versus default imagery in educational and hiring contexts [21, 42].

In closing, this work reframes generative imagery as an engineering concern with measurable societal impact. The results underscore that representational defaults are neither neutral nor inevitable, but design choices that can be audited, governed, and improved. By releasing a complete anonymized artifact package and positioning bias audits as routine practice, we aim to support a shift from documenting harms toward building accountable, inclusive, and AI-aware software engineering systems. The question is no longer whether generative models shape professional identity, but whether software engineering will take responsibility for how they do so.

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
