# OpenReview forum: "Auditing Who Appears to Belong: A Large-Scale Empirical Study of Bias in Deployed Text-to-Image Systems for Software Engineering"
_ACM.org/AIWare/2026/Conference — AIware 2026_

### Official Review · Reviewer_2sDH · 2026-02-20

**Rating:** 2
**Confidence:** 4

**Review:**

Strengths:

Clarity: the presentation is clear with background properly presented.

Quality: the experimental design and the evaluations for the proposed research questions are technically sound. The annotation is of high quality.

Originality and significance of this work: while existing work has identified gender, age, and race bias in image generation models [22,28], this paper further confirmed that such bias is intensified by AI models when compared to actual occupational reference statistics. This paper also analyze the impact of Roles, Contexts, and Prompts. The significance is moderate.

Weaknesses:

1. The authors proposed to integrate bias audits into engineering workflows with the prompt suite acting as the “representational regression tests.” However,  based on the discussion on annotation in Section 3.2, manual annotations are still required for the regression tests. This is not feasible since regression tests should be executed frequently so it is better automated.

2. No solution is discussed or proposed for the bias findings. The contribution could be more significant if the following RQs were explored:

    2.1. Whether this problem can be easily fixed by adding one requirement in the prompt such as "Please generate a diverse representation in race, gender, and age."

    2.2 Whether the models can generate the image correctly when directly prompted with "“Generate a realistic image of a black female software engineer of Age 52." Because of the imbalance in the training data, it is also possible that the quality of the generated images for the underrepresented groups is lower than that of the dominating groups.

**Summary:**

The paper empirically audit four text-to-image models in generating "software engineer" images. The generated images were annotated by both human and LLMs for demographic representation and portrayal cues. The study found that the generated images were highly skewed -- representing much more young white male characters than the actual occupational reference statistics. In addition, by varying the role, seniority, team context,  geography, and language, correlations between the role and age, geography and race were identified by gender bias still exists.

---

> ### Author Response · Authors · 2026-03-17
>
> We thank the reviewer for the positive assessment of the experimental design and annotation quality, and for highlighting the practical implications of integrating bias audits into engineering workflows. We also appreciate the reviewer’s supportive evaluation of the paper.
>
> We agree that mitigation techniques are an important and valuable research direction. The primary goal of this work, however, is to establish an empirical audit protocol and to quantify representational patterns across deployed text-to-image systems. Evaluating mitigation strategies; such as diversity-aware prompting, sampling constraints, or re-ranking mechanisms, would require controlled experimental setups and model interventions that fall outside the scope of the present audit.
>
> That said, we fully agree that such directions are important next steps. In the camera-ready version, we will expand the discussion to more explicitly highlight how the dataset, audit protocol, and analysis framework introduced in this paper can serve as a foundation for systematically evaluating mitigation strategies in future work.
>
> Thank you again for the positive decision and constructive feedback.

---

### Official Review · Reviewer_4gP6 · 2026-03-11

**Rating:** 3
**Confidence:** 3

**Review:**

+ Positioned at the crossroads of AI-aware and Software Engineering, the paper proposes very interesting contributions that are not only empirical but also methodological. In this regard, the authors demonstrated that representational bias in generative imagery is measurable, reproducible, and stable across model ecosystems, and therefore should be treated as an auditable quality attribute rather than an incidental artifact. By combining large-scale image generation, mixed methods analysis, and workforce-referenced diagnostics, their study  proposes a concrete audit protocol that can be integrated into Software Engineering workflows concerned with accountability, governance, and responsible AI adoption.
+The study is topical and novel
+The paper is well-written, clear, and concise.
+ The intersectionality of the analyzed dimensions (e.g., seniority levels and geographic location) is explored in the empirical study. This allows a better understanding of the compounding effects of the investigated dimensions on the generated results.
+The authors released a replication package containing prompts, scripts, datasets, generated images, coding schema, and inter-rater agreement statistics. This enhances the transparency and the reproducibility of their empirical study.

- The authors stated that they generated 880 images using 22 prompts across four models (ten images per prompt). However, the rationale for selecting these four models is not clearly explained. In addition, it is unclear how the 22 prompts were generated. Besides, the prompting strategies used to create the 22 prompts are not discussed in the paper. Furthermore, the total number of generated images (880) is relatively small, which raises concerns about the scalability of the study. As a consequence, the study findings may not generalize well to other models, prompting strategies, or broader image-generation scenarios.
-Generative AI models like the ones analyzed in the study are usually prone to hallucinations (e.g., they may generate non-sensical images such as people with extra fingers or distorted hands). However, the authors never discussed the potential impact of these hallucinations on the results and how they tried to minimize these hallucinations (if any).
-The models used in the study are known to be non-deterministic, so their outputs may vary from one execution to another when prompted with the same input. However, the authors never discussed this non-determinism in their work and did not explain how they tried to address it.
-Although there is a replication package, the configuration of the analyzed models shall have been briefly discussed in the paper. The authors should refer to Section IV.C of the paper by Wagner et al. (see reference below) to know how to describe such configurations.
Wagner, S., Barón, M. M., Falessi, D., & Baltes, S. (2025, May). Towards evaluation guidelines for empirical studies involving llms. In 2025 IEEE/ACM International Workshop on Methodological Issues with Empirical Studies in Software Engineering (WSESE) (pp. 24-27). IEEE.
-To assess linguistic effects on representation bias, the authors also investigated non-English prompts and decided to only focus on two non-English languages, namely: Spanish and Arabic. Their study design deliberately excludes non-English prompts written in other widespread languages such as mandarin, Swahili, and Hindi. This does not allow a correct comparison against regional labor statistics or the validation of the global persistence of role stereotypes.
-The study investigates representation bias in existing models by focusing on several dimensions (e.g., seniority levels and geographic location), but it does not focus on very critical dimensions such as disabilities and/or neurodivergence. So, the study misses the opportunity to provide a more comprehensive assessment of representation bias across important dimensions such as disabilities and neurodivergence.

Minor comment:
-Some figures are not readable i.e., are blurred (e.g., Figure 1, Figure 2, Figure 3).

**Summary:**

This study shows that contemporary text-to-image systems do more than mirror existing workforce imbalances. Across models, prompts, and contexts, they consistently produce a constrained visual narrative of the software engineering workforce that centers on youth, masculinity, and narrowly defined professional stereotypes. As these systems become embedded in educational materials, recruitment pipelines, documentation, and public communication, they operate as implicit socio-technical actors that shape who is “perceived” as belonging to the Software Engineering landscape.

---

> ### Author Response · Authors · 2026-03-17
>
> We really thank the reviewer for the detailed and constructive feedback and for recognizing the methodological contribution of positioning representational bias as an auditable property of AI systems in software engineering contexts. We are particularly encouraged by the decision, and we thank you for the suggested improvements:
>
> -Model selection: The four models were selected to represent distinct deployment ecosystems: a commercial system with explicit safety interventions (GPT-4o/DALL·E 3), a large proprietary platform model (Llama-4/Emu), a regionally trained multimodal model (Qwen3-235B-A22B), and an open-source diffusion model (Stable Diffusion). This combination enables comparison across different training corpora, mitigation strategies, and deployment contexts.
>
> -Prompt design and generation rationale: The 22 prompts were derived from prior literature on occupational stereotypes and software engineering roles and were designed to cover multiple dimensions where representational bias may emerge (roles, seniority, workplace contexts, geographic cues, and language). Prompts were intentionally phrased in a demographically neutral format to capture default model associations rather than responses conditioned by demographic instructions.
>
> -Dataset size and scalability: The dataset size reflects a trade-off between scale and rigorous human annotation. Each image is annotated across multiple demographic and portrayal dimensions using a mixed-methods coding protocol. While larger automated audits are possible, the present design emphasizes annotation reliability and interpretability.
>
> -Hallucinations and non-determinism: Generative artifacts and stylistic anomalies were intentionally retained to avoid introducing selection bias. Regarding non-determinism, generating multiple images per prompt captures stochastic variability while maintaining controlled prompt conditions. We will clarify these design choices in the methodology.
>
> -Model configuration reporting: We appreciate the suggestion to clarify model configurations. In the camera-ready version we will provide additional details regarding model access interfaces, default parameters, and generation conditions to align with emerging guidelines for empirical studies involving generative models.
>
> -Language scope and additional dimensions: The two non-English prompts were selected to provide an initial probe of multilingual effects across linguistically distinct contexts. We agree that expanding cross-linguistic analysis and additional demographic dimensions such as disability representation would be valuable extensions and will clarify these as future research directions.
>
> -Figures: We will provide higher-resolution versions of the figures in the camera-ready version to improve readability.
>
> We thank you again for your decision and the constructive feedback you provided to us.

---

### Official Review · Reviewer_5nqe · 2026-03-11

**Rating:** 3
**Confidence:** 3

**Review:**

### Strengths

**S1. Timely and well-positioned topic.** As generative imagery becomes embedded in SE artifacts, slides, documentation, recruiting materials, auditing representational bias is directly relevant to AIWare's mission. The paper convincingly frames visual bias as an SE quality attribute rather than a peripheral ethics concern, bridging fairness research and software engineering practice.

**S2. Systematic and reproducible methodology.** The four-component design (image generation, annotation/coding, quantitative analysis, qualitative thematic analysis) is well-structured. The 22 prompts span five meaningful dimensions, and benchmarking against workforce reference statistics elevates the analysis beyond purely descriptive auditing. The released replication package (prompts, scripts, datasets, coding schema, inter-rater statistics) strengthens reproducibility.

**S3. Meaningful cross-model comparison.** Including four models from distinct ecosystems, commercial U.S. (GPT-4o), commercial China (Qwen), Meta (Llama-4), and open-source (Stable Diffusion), enables comparative analysis across mitigation philosophies and training regimes. The finding that model differences are "of degree rather than direction" is an empirically grounded and practically significant observation.

 ### Weaknesses

**W1. Sparse intersectional cells.** With 880 images total, many intersectional categories contain near-zero counts (e.g., 3 women aged 40+, 0 older women of color). While aggregate patterns are statistically robust ($\chi^2$ tests, $p < 10^{-5}$), intersectional claims such as "older women of color are virtually absent" rest on cell counts too small for reliable inference. The authors should discuss minimum cell-size requirements explicitly and consider exact tests or bootstrap methods to quantify uncertainty at finer granularity.

**W2. Ecological validity of prompt design.** All 22 prompts follow a uniform template ("Generate a realistic image of a [role]"). Real-world SE usage of T2I models likely involves more varied and contextual prompts (e.g., "create an illustration for our team page"). Sampling actual prompts from SE practitioners or analyzing how T2I tools are invoked in practice would strengthen external validity. The current design, while internally consistent, may not fully capture how bias manifests under naturalistic conditions.

**W3. Insufficient validation of automated annotation.** DeepFace and GPT-4o serve as automated cross-checks, but per-category agreement rates between human and automated labels are not reported. Given documented biases in facial analysis tools, particularly for darker skin tones and non-Western faces, this triangulation could introduce systematic error. A confusion matrix or stratified agreement analysis would substantially strengthen the methodological contribution.

**W4. Statistical rigor of portrayal analysis (RQ4).** While the demographic analysis (RQ1–RQ3) employs appropriate statistical tests, the portrayal findings (RQ4) are largely descriptive. Claims such as "male-coded images include more visible props on average (2.91) than female-coded images (2.35)" lack significance tests, effect sizes, or confidence intervals. Given the severe class imbalance (843 male vs. 34 female images), such comparisons require careful statistical treatment to be interpretable.

**Summary:**

This paper conducts a mixed-methods empirical audit of 880 images generated by four text-to-image (T2I) models—GPT-4o/DALL·E 3, Llama-4/Emu, Qwen3-235B-A22B, and Stable Diffusion, using 22 demographically neutral prompts targeting software engineering roles varying by specialization, seniority, team context, geography, and language. Human annotations, triangulated with automated raters (DeepFace, GPT-4o), capture demographic representation (gender, race/ethnicity, age) and portrayal cues (setting, attire, props, emotion). Results are benchmarked against occupational workforce statistics. The central finding is that all four models converge on a narrow archetype, young, male, predominantly white, with women, older professionals, and several racial/ethnic groups systematically underrepresented. The authors propose actionable implications for AI-aware SE practice, including representational regression tests in CI/CD pipelines and diversity-aware generation defaults.

---

> ### Author Response · Authors · 2026-03-17
>
> We really thank the reviewer for the careful reading of the manuscript and for recognizing the relevance of framing representational bias as a quality attribute in AI-aware software engineering workflows. We are encouraged by your decision and we appreciate the suggestions regarding statistical reporting and methodological clarity.
>
> -Intersectional sample sizes: We agree that some intersectional categories are sparse (e.g., older women of color). In the manuscript, these observations are presented primarily as descriptive patterns rather than inferential claims. The principal statistical conclusions of the study rely on aggregate demographic distributions (gender, race/ethnicity, age) where sample sizes are sufficient and statistical tests remain robust. To further strengthen transparency, in the camera-ready version we will explicitly note minimum cell-size considerations and add bootstrap confidence intervals or exact tests for selected intersectional combinations.
>
> -Prompt ecological validity: The uniform prompt structure was intentionally designed to isolate default representational priors of the models. By removing demographic cues and keeping prompt wording constant, the study minimizes confounding effects introduced by prompt engineering choices and enables clearer attribution of representational patterns to model behavior. We agree that real-world prompts may be more varied; accordingly, we will clarify that the current design represents a controlled audit baseline, while more naturalistic prompt scenarios represent a valuable extension for future work.
>
> -Automated annotation validation: Automated tools (DeepFace and GPT-4o) were used only as cross-checks, while final labels rely on human consensus annotation. Inter-rater reliability between the two human coders is already high (e.g., κ≈1.0 for gender and κ≈0.83 for race/ethnicity). We agree that reporting agreement statistics between human and automated labels would further strengthen methodological transparency, and we will include a confusion matrix and stratified agreement rates in the camera-ready version.
>
> -Statistical treatment of portrayal analysis (RQ4): We appreciate the suggestion to provide statistical comparisons for portrayal metrics. The portrayal analysis was intended as a mixed-methods complement to the demographic audit; however, we agree that additional statistical reporting would improve interpretability. In the camera-ready version we will include appropriate statistical comparisons and effect-size reporting for selected variables such as prop counts and device presence across demographic groups.

---

### Author Response · Authors · 2026-03-17

We would like to thank all reviewers for the careful reading of the paper and for the constructive feedback. We are particularly glad to see that the reviewers found the topic timely and well aligned with the goals of AI-aware software engineering, and that the overall empirical design; combining controlled prompt generation, mixed-methods annotation, and cross-model comparison, was considered methodologically sound. We also appreciate the recognition that the work aims not only to document empirical patterns but to introduce a reproducible audit protocol that can be practically integrated into software engineering workflows.

In the responses for every reviewer, we address the specific points raised and clarify several methodological aspects of the study. In particular, we provide additional explanation regarding the rationale behind model selection and prompt design, how stochastic model behavior was handled, and how automated annotation signals were used strictly as triangulation rather than replacing human coding. We also clarify how intersectional results should be interpreted given the sparsity of some categories. Several of the reviewers’ suggestions; such as reporting additional agreement statistics, clarifying model configuration details, and improving the readability of figures, are helpful improvements that we will incorporate in the camera-ready version.

Overall, we are encouraged that the reviewers largely agree on the core contribution of the paper: that representational patterns in generative imagery for software engineering can be systematically measured and audited, and that these patterns are surprisingly consistent across different model ecosystems. We hope that the clarifications provided help address the remaining questions and further strengthen confidence in the study and its conclusions.